# Immunomodulatory Effects of Endodontic Sealers: A Systematic Review

**DOI:** 10.3390/dj11020054

**Published:** 2023-02-17

**Authors:** Jindong Guo, Ove A. Peters, Sepanta Hosseinpour

**Affiliations:** School of Dentistry, The University of Queensland, Herston, QLD 4006, Australia

**Keywords:** endodontic materials, sealer, immunomodulation, biocompatibility, regenerative endodontics

## Abstract

Inflammation is a crucial step prior to healing, and the regulatory effects of endodontic materials on the immune response can influence tissue repair. This review aimed to answer whether endodontic sealers can modulate the immune cells and inflammation. An electronic search in Scopus, Web of Science, PubMed, and Google Scholar databases were performed. This systematic review was mainly based on PRISMA guidelines, and the risk of bias was evaluated by SYRCLEs and the Modified CONSORT checklist for in vivo and in vitro studies, respectively. In total, 28 articles: 22 in vitro studies, and six in vivo studies were included in this systematic review. AH Plus and AH 26 can down-regulate iNOS mRNA, while S-PRG sealers can down-regulate p65 of NF-κB pathways to inhibit the production of TNF-α, IL-1, and IL-6. In vitro and in vivo studies suggested that various endodontic sealers exhibited immunomodulatory impact in macrophages polarization and inflammatory cytokine production, which could promote healing, tissue repair, and inhibit inflammation. Since the paradigm change from immune inert biomaterials to bioactive materials, endodontic materials, particularly sealers, are required to have modulatory effects in clinical conditions. New generations of endodontic sealers could hamper detrimental inflammatory responses and maintain periodontal tissue, which represent a breakthrough in biocompatibility and functionality of endodontic biomaterials.

## 1. Introduction

The inflammatory immune response is the initial reaction against physical and chemical stimulus or pathogens and is strictly regulated by the immune system [1]. This response is typically classified into two groups: acute inflammation and chronic inflammation. Acute inflammation commonly starts with the proliferation of monocytes and neutrophils, which can scavenge necrotic and damage cells by phagocytosis which ultimately leads to control of infection. At the early stage of inflammation, the resident macrophages (M1 phenotype) activate and secrete the proinflammatory cytokine, including interleukin (IL)-1, IL-6, and tumor necrosis factor alpha (TNF-α), to amplify inflammation [2]. The inflammatory pathways encompass various signaling pathways, including mitogen-activated protein kinases (MAPKs) and nuclear factor-κB (NF-κB) pathways, toll-like receptors (TLRs) pathway, cyclooxygenase-2 (COX-2), and inducible NO synthase (iNOS) related signal pathways [1]. Immune responses induced by sealers are complex and several studies reported endodontic sealers, such as epoxy resin-based and glass-ionomer sealers, had immunomodulatory impacts via various signal pathways [3,4,5]. Bioceramics-based sealers, containing mineral trioxide aggregate (MTA) or other calcium silicate materials, have been widely used in endodontic sealers due to their superior physicochemical and mechanical properties [6]. A study demonstrated that calcium silicate-based sealers, MTA Fillapex and EndoSequence BC, down-regulated pro-inflammatory cytokines, such as TNF-α and IL-6, to inhibit inflammation response and promote osteogenic differentiation in MC3T3-E1 cell lines in an inflammatory environment elicited by lipopolysaccharide (LPS) stimulation [7].

Tissue repair is closely correlated with inflammation and immune reaction [8]. The tissue repair process starts with the granulation-tissue formation and collagen accumulation. Macrophages contribute to the healing process and modulation of the immune system—specifically, the transition from the pro-inflammatory M1 phenotype to the anti-inflammatory M2 phenotype [9]. As a result, pro-inflammatory neutrophils are phagocytosed by the M2 phenotype. Cytokines, such as IL-10 and transforming growth factor beta (TGF-β), are released by M2-like phenotypes to induce inflammation resolution and stimulate the proliferation of fibroblasts cells to promote wound healing [10,11]. The coagulation cascade induced by platelet proliferation results in fibrin mesh formation and modulates polyphosphates and prothrombin to form thrombus to prevent bleeding and fluid loss in the early stage of healing [12]. Wound macrophages in granulation tissue can convert into fibroblast cells and produce collagen [13]. Fibroblast migration and collagen deposition also can eventually accelerate the healing process [14].

Pulpal and periapical lesions are inflammatory responses subsequent to microbial ingress and establishment in the root canal system [15]. For example, periapical granuloma is the most common histological correlate to asymptomatic apical periodontitis, a chronic inflammation response commonly caused by root canal infection and necrosis of the dental pulp [16]. For endodontic treatment, the emergence or persistence of inflammation is criteria for long-term success or failure. The development and regression of inflammation is involved in proliferation and recruitment of inflammatory cells, vascularization, and tissue healing. Endodontic sealers commonly as a foreign body react with periapical tissues and induce inflammation [17]. Intraosseous dental implanted AH Plus and EndoREZ exhibited severe acute inflammation responses with infiltration of leucocytes and macrophages and the increase of necrotic bone fragments [18]. However, acute inflammation may convert into chronic inflammation and failure to heal with long-term exposure to a stimulus [19]. The minority of periapical granuloma could further progress to periapical cysts that may not heal due to the persistence of inflammation stimulation [20]. For example, long-term stimulation of AH Plus induced chronic inflammation with the presence of lymphocyte and plasma cells [21]. In order to promote a successful endodontic treatment, endodontic sealers are required to be biocompatible, inert, or bioactive to modulate a favorable immune response for healing and have adequate physical properties. Bioactivity of the endodontic sealers attracted lots of attention recently due to the fact that they are directly in contact with the cells on the surface of the roots via accessory canals and through the apical foramen.

Biomaterials were defined as organic or inorganic materials used to replace, repair, or enhance disrupted or lost tissue [22]. From the function of material in the biological environment, biomaterials can be divided into bioinert and bioactive, biostable, biodegradable materials. As implants, early biomaterials were mainly inert and biocompatible, such as titanium and titanium alloy, which have long been used as dental implants [23]. The emergency of biodegradable materials is to mitigate the long-term influence of biomaterials existing in bodies and to avoid secondary surgeries with material aging and the repairment of damaged tissues [24]. For example, polylactic acid that was used in bone repair can be degraded to lactic acid and glycolic acid and eventually excreted [25]. Furthermore, platelet-rich fibrin derived from an individually autologous source is a biodegradable material, which could release cytokines to regulate inflammation and promote healing of pulp tissues [26]. Biomaterials with immunomodulatory functions have shown therapeutic effects to modulate inflammation and treat autoimmune diseases via regulating immune cells [27,28].

An endodontic sealer is a significant material to fill the space between the root canal wall and gutta-percha cones in root canal treatment, which spans various properties and compositions [29]. An ideal endodontic sealer requires various characteristics, including superior radiopacity and sealing ability, low solubility and viscosity, suitable setting time, non-cytotoxicity, adhesion to the dentinal wall to prevent leakage, and other physicochemical properties. Based on the chemical composition, current endodontic sealers can be divided into five groups: resin-based endodontic sealers, bioceramic-based sealers, zinc oxide-eugenol (ZOE)-based sealers, calcium hydroxide-based sealers, and glass ionomer sealers [30,31]. Recently, several studies reported endodontic sealers had immunomodulatory effects on inflammation and osteogenesis [32,33]. This immunomodulatory effect can influence the behavior of immune cells and regulate the release of chemokines and cytokines via various immunological pathways. Root canal sealers (endodontic materials) commonly up-regulate inflammatory cytokines, such as IL-6, IL-8, IL-12, and TNF-α, in early stage of inflammation, which result in inhibition of cell growth and high cytotoxicity [34]. However, endodontic sealers can also down-regulate the inflammatory cytokines, including IL-6 and TNF-α, induced by LPS, exhibiting a positive immunomodulatory effect [5]. In addition, endodontic sealers can also promote fibroblast proliferation and tissue regeneration [35]. Moreover, calcium silicate-based sealers, such as MTA Fillapex, can also promote osteogenic differentiation ability and calcium nodule formation, which may reduce bone resorption caused by inflammation [7]. Therefore, endodontic sealers with immunomodulatory effects could be a promising strategy for promoting the healing and tissue regeneration process for regenerative medicine.

In summary, it has been shown that biomaterials can regulate inflammatory responses and contribute to healing in vitro. However, the immunomodulatory impacts of endodontic sealers have not been comprehensively described. This systematic review aims to investigate the immunomodulatory properties of various endodontic sealers.

## 2. Materials and Methods

The Preferred Reporting Items for Systematic Reviews and Meta-Analyses (PRISMA) statement guideline [36] was followed in this systematic review. Study types included were in vitro and in vivo studies which investigated the inflammatory immunomodulation of various types of endodontic sealers.

### 2.1. Study Selection

(1)Studies must relate to “immunomodulatory effect of sealer” or “pro-inflammatory effect of sealer” or “anti-inflammatory effect of sealer” or “tissue regenerative ability”.(2)Studies must use proper and quantitative methods, such RT-PCR and ELISA to investigate the potential immunomodulatory effects of sealers.(3)The main results of studies must relate to at least one of these keywords: “macrophages”, “cytokines”, “immune cells”, “inflammatory cells”, “immunomodulation”, and “inflammation” or “anti-inflammation”.(4)Studies published in press within the last 15 years were included to obtain the most recent evidence.(5)Studies do not investigate the interaction between sealers and immune cells in the inflammation process or tissue repairment process.

### 2.2. Search Strategy and Information Sources

A systematic electronic search was conducted in the Scopus database from 2007 to 2022 and additional articles from the Web of Science, Google Scholar, and PubMed databases. This search was only limited to English language publications with the full text available, and original articles (not review articles, letters, or book chapters). The published papers were based on keywords, titles, and abstracts searched using the following query: (sealer AND regeneration) OR (sealer AND inflammation) OR (sealer AND cytokine) OR (sealer AND immunomodulatory) OR (sealer AND macrophage) OR (sealer AND healing) OR (sealer AND heal) OR (sealer AND modulation) OR (sealer AND anti-inflammatory). The flow chart presented summarizes the search process, shown in Figure 1.

### 2.3. Selection Procedure

The search results were exported in Research Information Systems (RIS) format from each database into reference manager software (EndNote 20 desktop version, Clarivate, Philadelphia, PA, USA). In the initial search, titles and abstracts were reviewed based on the inclusion criteria and duplicate results were discarded by hand. Full texts of all included papers were retrieved and reviewed.

### 2.4. Data Items

Data were summarized according to the following variables: (1) Authors and year of publication; (2) study type (in vivo or in vitro); (3) cell samples or animal model; (4) endodontic sealers; (5) types of markers; (6) methods to evaluate immunomodulatory effects; (7) outcomes of studies (immunomodulatory effects).

### 2.5. Risk of Bias Assessment

The ‘Modified CONSORT checklist of items for reporting in vitro studies of dental materials’ was used to evaluate the quality assessment in vitro. The RoB (Risk of Bias) tool for animal intervention studies ‘SYRCLEs RoB tool’ was utilized to assess the methodological quality assessment of the in vivo study [37,38].

## 3. Results

As shown in (Figure 1), a total of 28 articles are included in this review, including 22 in vitro and six in vivo studies from 2007 to 2022. After full-text search, 26 articles were excluded, and the reasons were shown as the following: Reason 1: Studies evaluated the cytotoxicity rather than the immunomodulatory effect of sealers (n = 15), Reason 2: Studies only evaluated the osteogenic capacity (n = 4), Reason 3: Study materials only contain MTA or endodontic cement (n = 4), Reason 4: Studies evaluated different endodontic treatments and apical healing (n = 2), Reason 5: Studies only evaluated the antibacterial effect (n = 1). All the included articles investigated the immunomodulatory effect of endodontic sealers by up-regulating or down-regulating inflammation-related molecules. The characteristics of in vitro and in vivo studies were summarized according to cell types or animal models, endodontic sealers, markers, methods, and immunomodulatory effect.

### 3.1. In Vitro Studies

The immunomodulatory effects of endodontic sealers from 22 in vitro studies were summarized in (Table 1). In terms of the cell model in the inflammatory assessment, included articles used several human primary cells, such as human dental pulp stem cells (hDPSCs), PDLSCs, osteoblasts and fibroblasts, human gingival fibroblasts, and mononuclear cells. Furthermore, animal primary cells included morphologic characteristics of macrophages, bone marrow-derived macrophages (BMDM), and M1 and M2 peritoneal inflammatory macrophages. The human cell lines used included u937 macrophages, THP-l monocytes, MG-63 osteosarcoma cells, immortalized human dental pulp stem cells (IHDPSCs), and immortalized mouse bone marrow monocytes (IMBMMs). The animal cell line included RAW 264.7 macrophages, MC3T3-E1 cells J774.1 murine macrophages, and L929 mouse fibroblast cells. In addition, hPDLSCs and RAW 264.7 macrophages were the most common cell models to evaluate endodontic sealers’ immunomodulatory effect.

Endodontic sealers included in this review were divided into five groups: (1) Epoxy resin-based endodontic sealers, including AH26 and AH plus, Epiphany (EPH), MetaSEAL and Endo-Rez (ER); (2) Zinc oxide-eugenol-based sealers, including N2 Universal, Endofill, and Pulp Canal Sealer; (3) Calcium hydroxide-based materials, including Sealapex, Sealapex Xpress, and Apexit Plus; (4) calcium silicate- based sealers, including MTA Fillapex, Bio-C Sealer, iRoot SP, BioRoot RCS, EndoSequence BC, Guttaflow Bioseal (containing both silicone and calcium silicate); and (5) Glass ionomer sealer, including surface-reaction-type pre-reacted glass-ionomer (S-PRG) filler containing root canal sealer (S-PRG sealer).

The cell viability after endodontic sealer treatment was mainly evaluated by the 3-[4,5-dimethylthiazol-2-yl]-2,5- diphenyltetrazolium bromide (MTT) assay, and only one study utilized the cell counting kit-8 assay and water-soluble tetrazolium (WST) salt assay. The cytokine production (interleukin and tumor necrosis factor), and other inflammation-related signal molecule, including COX-2, nitric oxide (NO) and iNOS, NO, and prostaglandin E2 were mainly evaluated by reverse transcription polymerase chain reaction (RT-PCR) or enzyme-linked immunoassay (ELISA) assay.

Figure 2 summarizes representative the ELISA results of the relative fold changes in the production of IL-6, IL-10, and TNF-α induced by three different endodontic sealers: MTA Fillapex, BioRoot RCS, and Pulp Canal Sealer in human periodontal ligament cells models with 0.2 mg/mL or 1:8 dilution of each sealer extracts. All entities were extracted and transferred to Prism (Version 9.0.0, GraphPad, La Jolla, CA, USA). Comparisons between groups were performed by a one-way or two-way analysis of variance (ANOVA), with post-hoc Tukey’s tests. A *p* value less than 0.05 was considered statistically significant.

MTA Fillapex elicited an increase in the production of both pro-inflammatory cytokines (TNF-α) and IL-6 and anti-inflammatory cytokines: IL-10, although there were no significant differences about the production of these cytokines (*p* > 0.05, ANOVA multiple comparisons). However, BioRoot RCS inhibited the production of pro-inflammatory cytokine IL-6 and promoted the expression of anti-inflammatory cytokine IL-10 (*p* < 0.0001, student’s *t*-test). Furthermore, PCS stimulated the production of IL-6 and inhibited the expression of TNF-α (*p* < 0.005, student’s *t*-test). Due to heterogeneity of each experiment, no comparable data in BioRoot RCS and PCS in terms of TNF-α and IL-10 secretion are shown in (Figure 2). MTA and similar calcium silicate-based materials, as well as zinc oxide-eugenol-based endodontic sealers showed anti-inflammatory effects in terms of IL-6, IL-10, and TNF-α production. 

It is difficult to compare the immunomodulatory effects in different endodontic sealers due to heterogeneity. Previous studies used different cell models, endodontic sealers, and markers. Furthermore, the method to prepare sealer extract is also different, but the biological evaluation standards ISO 10993-12 document was accepted by several recent studies to set different extract concentrations according to the property of material, such as state, size, thickness, and the extraction ratio [47,48]. Moreover, using extracts for in vitro biological testing can be clinically irrelevant [53].

Without LPS stimulation, either resin-based endodontic sealers (AH plus) or calcium silicate sealers (Endosequence BC Sealer and BioRoot RCS) displayed an increase in both pro-inflammatory (TNF-α) and anti-inflammatory cytokines (IL-6), which indicated these sealers induce an inflammation response initially as a foreign body and exhibit an anti-inflammatory effect on hPDLSCs. In addition, multimethacrylate-based sealers (Real Seal), calcium hydroxide-based sealers (Sealapex Xpress), and AH-plus also showed similar results in a SV40 T-Ag-transfected cell line of human pulp-derived cells [45,50]. A gradual reduction in the concentration of IL-6 and TNF-α were observed in hPDLSCs after sealer eluate treatment from 3 h to 24 h, which also indicated the resolution in acute inflammation (anti-inflammatory) effects of endodontic sealers in in vitro studies [51]. Moreover, with the LPS stimulation, endodontic sealers (AH Plus, MTA Fillapex and EndoSequence BC) inhibited the mRNA expression of TNF-α [7].

### 3.2. In Vivo Studies

Six in vivo studies were included, which investigated the subcutaneous tissue response in terms of endodontic sealer implants. Compared with in vitro studies, the main methods to investigate immunomodulatory effects of sealers were histology and immunohistochemical analysis.

Furthermore, five included articles used rats of different species or strains as animal models [54,55,56,57,58], while only one project used zebrafish [59]. In vivo studies mainly evaluated the macrophage infiltration, cytokine production such as IL-6 and VEGF and changes in fibrous capsule, and vascularization [55,56].

Resin-based sealers with different chemical composition displayed a tendency to elicit infiltration of specific immune cells in the initial inflammation responses, AH Plus preferentially induced MHC class II molecule-expressing cells and neutrophils, while MetaSEAL induced infiltration of macrophages first [54]. In addition, the in vivo studies showed that MTA Plus, MTA Fillapex, and AH Plus down-regulation of inflammation and formation of a fibrous capsule over time and the addition of petasin in zinc oxide eugenol sealer also displayed an anti-inflammatory effect [55,59]. EndoSequence BC Sealer HiFlow promoted M2-like macrophage polarization in vivo which indicates the anti-inflammatory effect [57]. MTA Fillapex displayed a significant suppression in IL-6 production compared to the AH Plus [55]. Another study showed that GuttaFlow Bioseal was less effective in secretion of VGEF and IL-6 than MTA Fillapex (superior anti-inflammatory effects) [55,56]. Table 2 summarizes the immunomodulatory effects of endodontic sealers in five in vivo studies.

### 3.3. In Vitro Studies

The risk of bias assessment is presented in the following Tables (Table 3 and Table 4). In terms of in vitro studies, 12 studies did not present a structured summary, while 10 articles used a structured abstract. Most of the research presented the *p* value instead of confidence interval, only a few studies reported the confidence interval or significance level α [4,40,42,48,49,50]. Furthermore, all included studies were cell-based and did not refer to implementation and intervention to teeth. However, four studies used the ISO 10993-12 standard to prepare sealer extracts [33,47,48,49].

With respect to in vivo studies, the age, weight, or gender were not presented completely [54,55,58]. None of the studies mentioned allocation concealment. Regarding random outcome assessment, two studies did not mention if the measurement and histological analysis were performed by blinded examiners, which could elicit a subjective error [54,59].

## 4. Discussion

### 4.1. Inflammatory Pathway and Signaling Mechanisms Related to Endodontic Sealers

Endodontic sealers have demonstrated significant modulation of cytokine production and are involved in various immunological pathways, including nuclear factor κB (NF-κB), MAPK, Janus kinase-signal transducer and activator of transcription (JAK-STAT) pathway, ROS related pathways [5,60,61,62,63,64]. Figure 3 summarizes endodontic sealers and modulated cytokines or other signal molecules production via various pathways mainly in macrophages and monocytes.

Epoxy–resin-based endodontic sealers, such as AH26 and AH Plus, activated stress-activated the protein/c-jun N-terminal kinase (SAPK/JNK) of MAPKs pathway, which regulates mitochondria-mediated apoptosis and survival and contributes to the cytotoxicity of these sealers [60]. Peroxisome proliferator-activated receptor (PPARγ), an anti-inflammatory transcription factor, which can suppress the SAPK/JNK pathway and block the translocation of NF-κB, was also inhibited by AH26 treatment in MC-3T3-E1 cells [61]. Furthermore, AH26-induced suppression of iNOS mRNA expression could down-regulate inflammatory responses [4]. The accumulation of ROS induced by oxidative stress can activate many transcriptional factors and involve in inflammatory pathways, such as MAPK and NF-kB pathways [65,66]. Based on this, polyphenols were used in endodontic sealers due to their antioxidants and anti-inflammatory effects in the arachidonic acid pathway of inflammation. ZOE-based sealers showed superior anti-inflammatory effects, such as inhibition of iNOS and COX-2 (Figure 3) [39,67]. Under normal condition, COX-2 induces not only inflammatory responses, but also regulates VEGF to promote wound healing [68]. However, overexpression of COX-2 contributes to the development and transition in oral mucosa, especially malignancies such as oral squamous cell carcinoma via the COX-2/PGE2 pathway [69]. Therefore, a potential synergistic effect of COX-2 expression in inflammatory stimulation should also be considered and local inhibition of COX-2 expression can be beneficial for endodontic sealers (Figure 3) [3,62]. Similarly, Hinokitiol, a natural plant component, was also added into Apexit Plus (calcium hydroxide-based sealer). This mixture showed a significant anti-inflammatory effect by preventing mRNA expression of COX-2, HIF-1α, and lipoxygenase (LOX) [44]. A recent study indicated that addition of simvastatin, a statin drug, can inhibit GTPases production in the mevalonate pathway and NF-κB pathway [52]. However, the relationship between simvastatin and VEGF depends on the cell type and drug concentration [70]. Therefore, the effect of simvastatin as a supplement of endodontic sealers requires further research to verify if a simvastatin supplemented sealer has anti-inflammatory and/or wound healing effects. MTA materials also activated the NF-κB and MAPK pathways to enhance the osteogenic capacity of periodontal ligament stem cells (PDLSC), which could be beneficial for periodontal tissue repairment and pulp regeneration [71]. A study showed that calcium silicate-based endodontic sealers, such as MTA Fillapex, exhibited impairment to inflammation and healing against bacterial infection in an early study by inhibiting the production of proinflammatory signal molecules: NO and ROS in M1 and M2 macrophages. Furthermore, tissue repairment-related cytokine IL-10 was also down-regulated by MTA sealers in the healing process [43]. However, calcium silicate-based BioRoot RCS RCSdown-regulated TNF-α and iNOS [50]. Findings of in vivo studies showed the resolution of inflammation and the accumulation of collagen after sealer treatment, which indicates that endodontic sealers promoted the healing process [55,57]. S-PRG sealers can release F, Sr, and Si ions, which enhance remineralization and may lead to restoration of dentine and reduced Zn ions can down-regulate pro-inflammatory cytokines [5,72]. Furthermore, S-PRG sealers exhibited an inhibition of transcription factor p65 (P65) of the NF-kB pathway, which may contribute in inflammation regression and promotion of healing (Figure 3) [5].

### 4.2. Modulatory Effects of Endodontic Sealers on Macrophages

Macrophages play an essential role in foreign body reaction and promote wound healing [9]. Although M1 macrophages can activate and enhance immune responses by secreting cytokines and chemokine, the healing process requires the anti-inflammatory M2 phenotype. The difference in the ratio of M1/M2 phenotypes also influences inflammation. For example, a higher M1/M2 ratio in periodontitis resulted in chronic periodontitis than gingivitis. [73]. Modulation of M1/M2 phenotypes could be beneficial for inflammatory immune regulation: the inhibition of the JNK pathway and the expression of RAC-beta serine/threonine-protein kinase (Akt2) can promote M2 phenotype polarization to attenuate inflammation and regulate the pro-inflammatory micro-environment [74]. 

The endodontic sealer overall showed alterations in macrophage polarization, cytokine production, and phagocytic activity (Figure 4). Calcium silicate-based sealers iRoot SP and mineral trioxide aggregate affected the expression of M1/M2 phenotype markers in RAW 264.7 [75]. Furthermore, a mixed macrophage pattern was also observed in mice bone marrow derived macrophages after treatment by AH Plus, Sealapex Xpress, and Endosequence BC Sealer [50]. In vivo studies demonstrated M2 macrophages involved in healing and tissue repair processes against tissue response to MTA materials and bioceramic materials enhanced the polarization of M2 macrophages [76,77]. Pro-root MTA cement has shown M2c polarization while iRoot SP induced M1 macrophages [46]. In vivo studies indicated that iRoot SP and EndoSequence BC Sealer HiFlow induced M2 macrophage polarization and exhibited inflammatory inhibition within 150 days [57]. Furthermore, the number of infiltrations of MHC class II molecule-expressing cells or macrophages were differently treated by AH Plus and MetaSEAL, which suggests the difference in immunogenic potential against a foreign body for different types of sealers. The immunomodulatory effects of endodontic sealers were also evaluated by other cytokines, especially interleukins and TNF-α. Eugenol in Tubli-seal depicted an inhibition of pro-inflammatory cytokines, such as IL-1β, IL-6, and IL-8 [33]. Another study indicated that hydrocortisone in Endomethasone modulated the production of IL-6 instead of eugenol [17]. Further research is required to assess the secretion of IL-1β and IL-8 or other cytokines in the presence of ZOE-based sealers. IL-6 as a pleiotropic cytokine plays an essential role in conversion from local acute inflammation to systemic inflammation [78]. However, IL-6 also showed beneficial impact on osteogenesis in DPSC. [79]. Hence, it is imperative to focus more on the signal transduction pathway of related inflammatory mechanisms induced by sealers for regenerative approaches.

### 4.3. Methodology in the Immunomodulatory Endodontic Sealers

Diverse methodology and variables made it impossible to reach conclusive decisions. In vitro studies used a different cell model to investigate immunomodulatory effects of sealers. Raw 264.7 macrophages are commonly used to evaluate inflammatory cytokines and other signal molecules releasing [3,4,5]. However, PDL cells could work as local immune cells and modulate inflammatory cytokine releasing. For example, LPS stimulation induced a rapid increasing in IL-6 production and toll-like receptors (TLRs) mRNA expression in PDL cells [80]. Unexpected leakage could contribute to a failure of endodontic treatment, and endodontic sealers could be exposed to periradicular tissue [30]. Hence, PDL cells and cells derived from periapical tissue could be valuable research material for inflammatory effects. Moreover, standardized three dimensional models could more accurately mimic PDLSC cells morphology, proliferation, migration, and immunomodulation under inflammation conditions. [47]. These novel methods are also more clinically relevant than two-dimensional culture and extracts. Furthermore, only a few studies used current biocompatibility tests for sealers extract preparation following ISO 10993 document [33,47,48,49]. The states of studied endodontic sealers were sometimes powder, such as Sealapex Xpress, BioRoot RCS, and Real Seal XT, which were made after setting and reported concentration of extracts solution as mg/mL [35,49]. This cannot be clinically relevant since the state of the sealer before and after setting has not been acknowledged. On the other hand, the concentrations of sealer extracts were reported as dilution ratio (such as 1:2, 1:4, and 1:8) [48] which can be misleading and obviously increased the level of difficulty of quantitative comparison amongst various studies.

### 4.4. Future Persective and Clinical Relevance

Modulation of the macrophage phenotype is a promising strategy for immunomodulatory biomaterials used in tissue repair [81]. Magnetic nanoparticles have displayed superior mechanical properties and promote osteogenic differentiation of cells via a magnetic effect [82]. Incorporation of magnetic nanoparticles improves the inflammation inhibitory effect of modified EndoREZ, which could promote apical periapical healing [83]. Furthermore, the incorporation of a substance with anti-inflammatory effect is also a convenient method to achieve a better anti-inflammatory property. For example, the addition of petasin in ZOE sealer exhibited an anti-inflammatory effect [84].

Although researchers have been optimizing biomaterials, especially in regenerative medicine, to decrease interactions with the immune system, the potential risks of unnecessary reactions are still a key issue that hinders their clinical application. In endodontics, it is even more challenging since the rationale of application and the definition of the favorable immunomodulatory effectare still unclear. In recent years, there are several studies trying to define an “ideal” material for root canal filling; however, there is no consensus about which material is best. Regarding the immunomodulatory effect, studies focused more on displaying the regenerative aspect of materials, such as osteoinduction and promotion of cementogenesis. Nevertheless, it seems the hard tissue regenerative aspect of immunomodulation is not clinically relevant based on application purposes, and this aspect is only easier to achieve and study ex vivo. Evidently, it is necessary to re-evaluate the rationale of immunomodulation depending on the material’s clinical application and purpose. Preservation of natural periodontal turnover, including periodontal ligaments, cementum, and bundle bone as well as hampering detrimental inflammatory responses are undeniably pivotal to address via the new generation of biomaterials.

## 5. Conclusions

Inflammation is a challenge for endodontic sealers. We found that endodontic sealers could modulate macrophages polarization and inflammatory cytokine production to promote healing, tissue repair, and inhibit inflammation. The immunomodulatory effects were mainly evaluated by the production of inflammation-related cytokines [7,17,40,42,45]. The new sealer entrants to the market display superior biocompatibility and more potential to modulate inflammation responses compared with older counterparts. Furthermore, different types of endodontic sealers play immunomodulatory roles in different models via various pathways. Although silicate-based sealers induced mild and transient cytotoxicity with an increase in pro-inflammatory cytokine TNF-α, this type of sealer exhibited a superior anti-inflammatory effect with less IL-6 production compared with AH Plus [48,51]. For instance, AH Plus and AH 26 down-regulated iNOS mRNA, while S-PRG sealers down-regulated p65 of NF-κB pathways to inhibit the production of TNF-α, IL-1, and IL-6. Compared with resin-based sealer Real Seal XT, calcium hydroxide-based Sealapex Xpress inhibited the production of TNF-α and could be beneficial for tissue regeneration [49]. In vivo studies also showed most endodontic sealers elicited initial inflammation with immune cell infiltration [54,55,58,59]. However, silicate-based endodontic sealers, such as GuttaFlow Bioseal and MTA Fillapex, exhibited superior suppression of chronic inflammation than zinc oxide-based sealer Endofill with the down-regulation IL-6 production and inflammatory cells. Furthermore, the down-regulation of IL-6 and VEGF could promote connective tissue repair [56]. Although modulatory effects of endodontic materials on the immune system have been shown by several researchers, transitioning from a detrimental crosstalk between biomaterials and a tissue-like foreign body reaction to a more efficient and intelligent tissue repair requires further investigation. Achieving a perfect functional integration in the host for endodontic materials means harmonic maintenance of periodontal tissue, allowing a natural turnover of various structures surrounding roots as well as avoiding potential complications, such as inflammatory resorptions. From a clinical point of view, this will represent a breakthrough in biocompatibility and functionality of endodontic biomaterials.

In conclusion, endodontic sealers have exhibited immunomodulatory effects on regulating cytokines release and influence macrophage phenotypes to inhibit inflammation, which could eventually promote healing. Further studies will contribute to delineate the mechanisms underlying the immunomodulatory effects of endodontic sealers and the relationship between endodontic sealers and healing.

## Figures and Tables

**Figure 1 dentistry-11-00054-f001:**
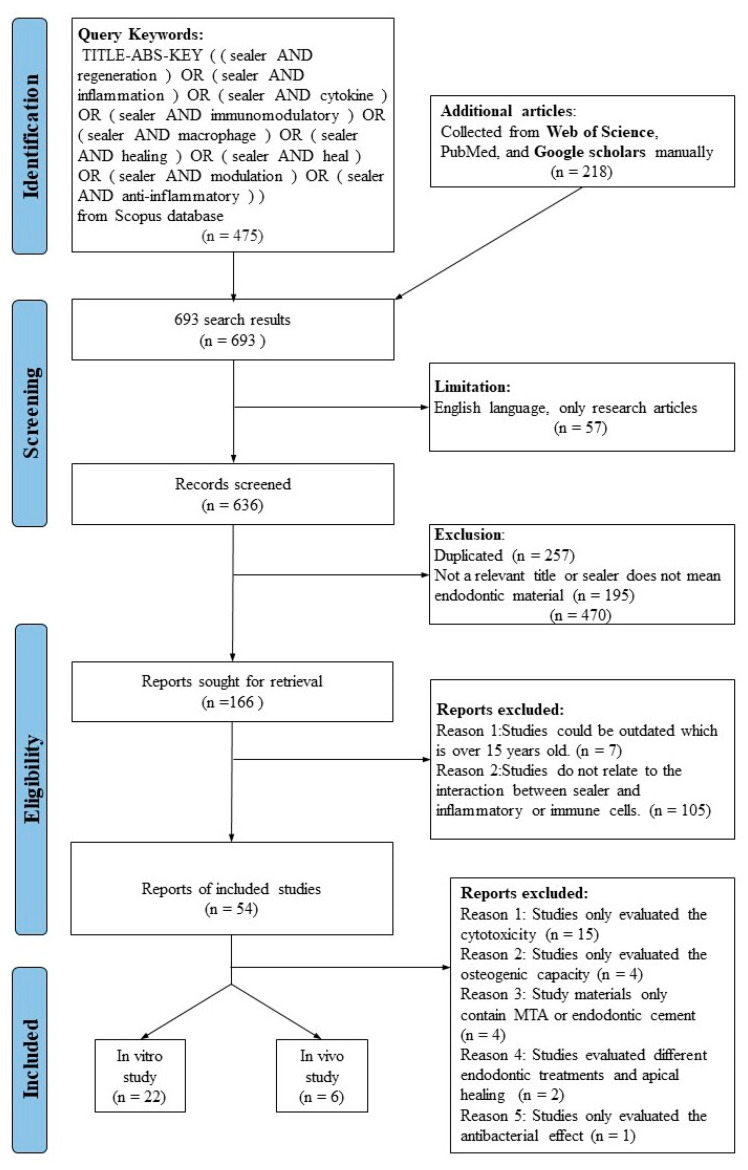
Study design based on PRISMA.

**Figure 2 dentistry-11-00054-f002:**
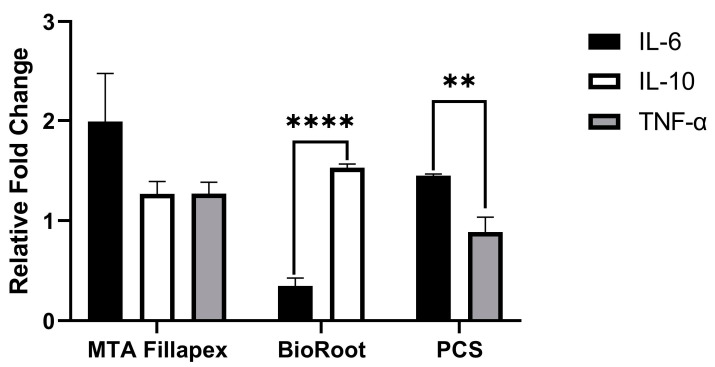
Relative fold change in cytokine secretion induced by different sealers. The relative fold changes in the production of IL-6 IL-10 and TNF-α were in red, yellow, and blue, respectively, based on data from [17,35,47,48]. Although MTA enhanced the production of IL-6, IL-10, and TNF-α, there were no significant difference (*p* > 0.05, ANOVA multiple comparisons). The relative fold changes in IL-6 and IL-10 elicited by BioRoot RCS (*p* < 0.0001, student’s *t*-test comparisons) and IL-6 and TNF-α (*p* < 0.005, student’s *t*-test comparisons) induced by PCS was significant. Abbreviation: **** BioRoot (BioRoot RCS) and ** PCS (Pulp Canal Sealer).

**Figure 3 dentistry-11-00054-f003:**
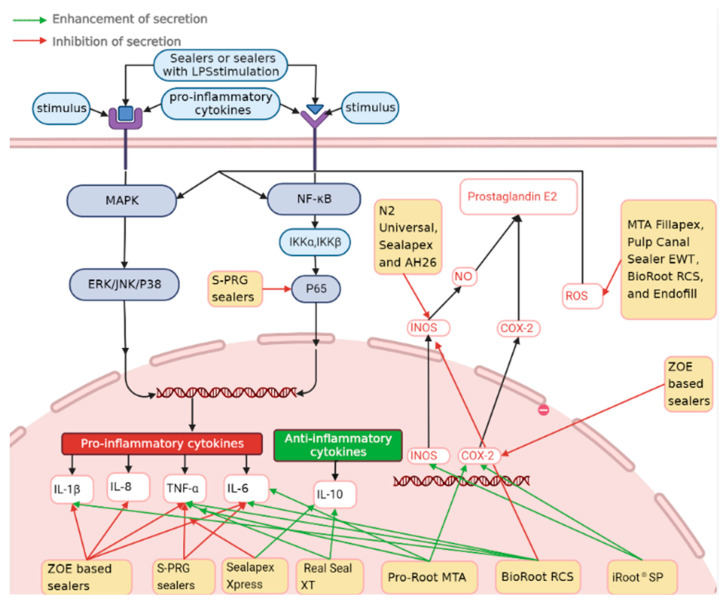
Immunomodulatory effects of endodontic sealers on cytokine production and potential pathways (Pro-Root MTA is an endodontic cement). Arrows in red and green represent the inhibition and enhancement effects of cytokines or signal molecules production on endodontic sealers in macrophages and monocytes.

**Figure 4 dentistry-11-00054-f004:**
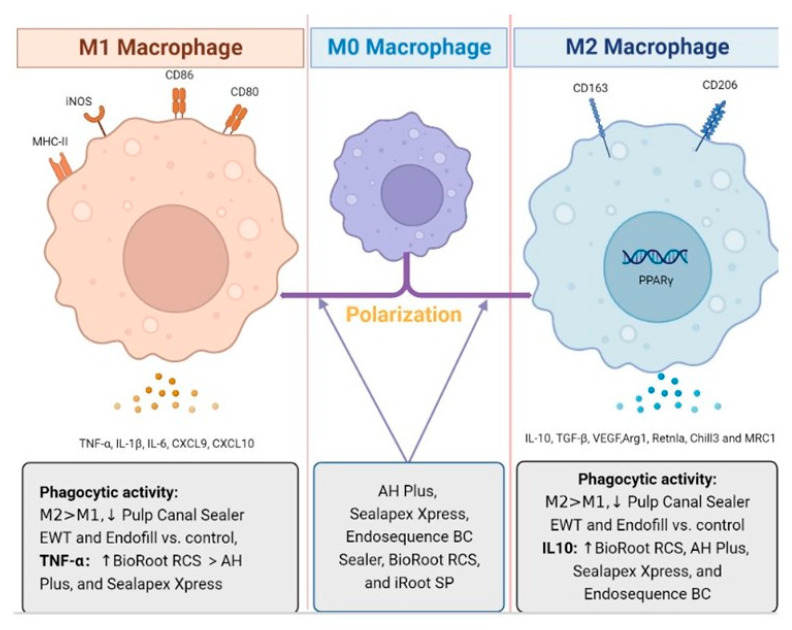
Modulatory effects of endodontic sealers on macrophage polarization, cytokine production, and phagocytic activity. AH Plus, Sealapex Xpress, Endosequence BC Sealer, BioRoot RCS, and iRoot SP promote both M1 and M2 polarization, while Pro-root MTA cement only promote M2 polarization. Pulp Canal Sealer EWT and Endofill decreased the phagocytic activity of both M1 and M2 macrophages, while M2 phenotypes have higher phagocytic activity than M1 macrophages. BioRoot RCS induced the highest TNF- α compared with BioRoot RCS, AH Plus, and Sealapex Xpress and all of them elicited IL-10, which is the polarization marker of M2 macrophages.

**Table 1 dentistry-11-00054-t001:** Summary of immunomodulatory effect of endodontic sealers in vitro studies.

Cell Type	Material	Marker	Method	Immunomodulatory Effect	Ref.
RAW 264.7 macrophages	AH26, Sealapex, and N2 Universal	COX-2	RT-PCR, agarose gel electrophoresis, Cell Counting Kit-8 assay	↑ COX-2 mRNA vs. control	[3]
RAW 264.7 macrophages	AH26, Sealapex, and N2 Universal	iNOS	MTT assay, RT-PCR, SDS-polyacrylamide gel electrophoresis, Colorimetric analysis	↓ iNOS mRNA vs. control	[4]
U937 macrophages	Zinc oxide eugenol sealers	IL-1β, TNF-α, PGE2, COX-2	ELISA, RT-PCR	↓ IL-1β, ↓TNF-α, ↓ PGE2, ↓ COX-2 mRNA vs. LPS	[39]
Human THP-l monocytes (ATCC TIB 202)	AH-Plus, Pulp Canal Sealer, Epiphany, Endo-Rez, and an experimental Endo-Rez	TNF-α, IL-1β, IL-6	ELISA, MTT assay	↓ IL-1β vs. control Inhibition: PCS > EPH > ER (α = 0.05) ↓ IL-6 vs. Control EPH > PCS (α = 0.05) ↓TNF-α vs. control EPH > PCS (α = 0.05)	[40]
Morphologic characteristics of macrophages from C57BL/6 mice	Pulp Canal Sealer EWT and Endofill	NO, ROS, TNF-α IL-10 IL-12	ELISA, Nitric Oxide Assay(colorimetric), ROS assay (Spectrophotometric assays)	↓ Phagocytic activity of macrophages ↓ ROS vs. control ↓ TNF-α when M2 cells + F. nucleatum + IFN-γ TNF-α: P. anaerobius + M1 cells + IFN-γ > M2 cells (*p* < 0.05)	[41]
THP1 human monocytic cells (ATCC TIB 202)	AH-Plus-Jet, Pulp Canal Sealer, MTA-type sealers, ProRoot White MTA, and an experimental calcium silicate-based sealer	IL-1α, IL-1β, IL-2, IL-3, IL-4, IL-5, IL-6, IL-7, IL-8, IL-10, IL-15, IFN-γ, TGF-β1, TNF-α, TNF-β, VEGF	Cytokine Array based on ELISA	↓ IL-2, IL-6, IL-10, and IL-15 ↑ IL-1α and IL-1β	[42]
M1 (from C57BL/6 mice) and M2 (from BALB/c mice) peritoneal inflammatory macrophages	MTA and MTA Fillapex (FLPX)	ROS, IL-12, IL-10, TNF-α, NO	Reactive Oxygen intermediates assay, MTT assay, Nitric Oxide Assay	↓ ROS and NO vs. control in both M1 and M2 cells. (*p* < 0.05) ↑ TNF-α: IFN-γ + FLPX + P. anaerobius stimulated M1 cells vs. control (*p* < 0.05) ↓ TNF-α: F. nucleatum-stimulated M2 cells vs. control (*p* < 0.05) IL-10: M1 > M2 cells	[43]
Human dental pulp and periodontal ligament stem cells, osteoblasts, and fibroblasts	EndoRez	IL-6, IL-8, IL-12, TNF-α	MTT assay, Fluorescence staining, confocal laser scanning microscope analysis, ELISA	↑ IL-6, IL-8, IL-12, and TNF-α: EndoRez vs. control	[34]
Human gingival fibroblasts, and Human osteosarcoma cell lines MG-63	AH Plus, Apexit Plus, and Canals	COX-2, HIF-1α, LOX	MTT assay, RT-PCR	↓ COX-2 and HIF-1α: Apexit Plus + 1% Hinokitiol vs. control in MG-63 (*p* < 0.05) ↓ COX-2, HIF-1α, and LOX: Apexit Plus + 1% Hinokitiol vs. control in HGF (*p* < 0.01, *p* < 0.001, *p* < 0.001, respectively)	[44]
Immortalized human dental pulp stem cells (IHDPSCs) and mouse bone marrow monocytes (IMBMMs)	Zinc oxide–eugenol (ZOE)-based endodontic sealers and cement (IRM and Tubli-Seal)	IL-1β, IL-6, IL-8, TNF-α	RT-PCR, WST assay, Live and Dead Cell Assay	↓ IL-1β, IL-6, and IL-8 vs. LPS control (*p* < 0.05) in HDPC ↓ IL-1β, IL-6, and IL-8 vs. control (*p* < 0.05) in HDPC ↑ IL-1β and IL-6: Zn 2+ vs. media-exchange condition (*p* < 0.05) in IMBMM ↓ TNF-α eugenol vs. control (*p* < 0.05) in IMBMM	[33]
SV40 T-Ag-transfected cell line of human pulp-derived cells	Apexit Plus, Real Seal, AH Plus, and EndoREZ	IL-6, IL-8, TNF-α	MTT assay, MTN assay, RT-PCR, Immunohistochemistry, ELISA	↑ IL6, IL8, and TNF-α vs. control (*p* < 0.05)	[45]
Primary human periodontal ligament (PDL) cells, human umbilical vein endothelial cells, and Inflammatory (THP-1) Cell	BioRoot RCS and Pulp Canal Sealer	IL-6, TGF-β1	Immunofluorescence, RT-PCR, ELISA, MTT assay	↓ IL-6 BioRoot RCS vs. control (*p* < 0.05) ↑ IL-6 PCS vs. control (*p* < 0.05) ↑ TGF-β1 BioRoot RCS vs. control (*p* < 0.05) ↓ TGF-β1 PCS vs. BioRoot RCS (*p* < 0.05) ↓ IL-6 BioRoot RCS vs. PCS (*p* < 0.05)	[35]
MC3T3-E1 cells	AH Plus, MTA Fillapex, and EndoSequence BC	IL-6, TNF-α, ALP, OCN	WST assay, RT-PCR, Alkaline Phosphatase Staining, Alizarin Red Staining	↓ IL6 and TNF-α vs. control (*p* < 0.05)	[7]
RAW 264.7 macrophages	Mineral trioxide aggregate (Pro-Root MTA, PR-MTA) and other calcium silicate-based materials (iRoot^®^ SP Injectable Root Canal Sealer, IR-BC)	TNF-α, IL-6, IL-1β, COX-2, iNOS	MTT assay, Western Blotting, RT-PCR	↑ iNOS iR-BC vs. control ↑ COX-2 iR-BC and PR-MTA vs. control ↑ TNF-α iR-BC and PR-MTA vs. control ↑ IL-1β iR-BC vs. control ↑ IL-6 R-BC and PR-MTA vs. control	[46]
Primary human periodontal ligament stem cells (PDLSCs)	BioRoot RCS, ProRoot ES, MTA Fillapex	IL-6, IL-8, GRO, IL-4, IL-10	Flow cytometry, MTT assay, Multiplex bead-based cytokine assay	↑ IL-6, IL-8, and GRO MTA Fillapex and AH Plus vs. control (*p* < 0.05) ↑ IL-4 and IL-10 BioRoot RCS vs. control (*p* < 0.05)	[47]
Primary human periodontal ligament cell	Endomethasone N (EN) and Pulp Canal Sealer (PCS)	IL-6, TNF-α	ELISA, spectrofluorimetry	↓ IL6 EN vs. control ↑ IL-6 PCS vs. control ↓ TNF-α EN vs. control ↑ TNF-α PCS vs. control ↑ IL-6 EN vs. PCS ↑ TNF-α EN vs. PCS (*p* < 0.05)	[17]
Primary hPDLSCs	Bio-C Sealer, MTA Fillapex, and PBS Cimmo HP	IL-10, TNF-α	ELISA, MTT assay, immunostaining, flow cytometry	↑ TNF-α Bio-C Sealer, Cimmo HP and MTA Fillapex vs. control (*p* < 0.05)	[48]
The J774.1 murine macrophage cell line	Sealapex Xpress and Seal Real XT	IL-4, IL-6, IL-10, TNF-α	ELISA, MTT assay	↑ TNF-α Real Seal XT vs. control (*p* < 0.05) ↑ IL-6 Real Seal XT vs. control (*p* < 0.05) ↓ TNF-α Sealapex Xpress vs. control (*p* < 0.05) ↑ IL-10 Sealapex Xpress vs. Real Seal XT (*p* < 0.05)	[49]
Bone marrow-derived macrophages (BMDM)	AH Plus, Sealapex Xpress, Endosequence BC Sealer, BioRoot RCS and Calen	GM-CSF, IL-10, IL-6, IL-1β, TNF-α M1 markers: Cxcl10, CxCL9 and iNOS M2 markers: Arg1, Retnla, Chill3 and MRC1	RT-PCR, MTT assay, Multiplex bead-based cytokine assay	Markers of M1 phenotype: ↓ iNOS BioRoot RCS and Sealapex Xpress vs. control (*p* < 0.001) Markers of M2 phenotype: ↓ Arg1 Sealapex Xpress vs. control (*p* < 0.05) ↓Retnla EndoSequence BC Seale vs. control (*p* < 0.001) Sealapex Xpress vs. control (*p* < 0.01) ↑ IL-1β, TNF-α, and IL-6	[50]
Human peripheral blood mononuclear cells (hPBMC), hPDLSCs	MTA Fillapex, BioRoot RCS, AH Plus, and Pulp Canal Sealer	IL-6, TNF-α, IL-8, IL-10	ELISA	↑ IL-6 MTA Fillapex > BioRoot RCS(*p* < 0.001) AH Plus > BioRoot RCS(*p* < 0.05) in hPBMC afer 12 h ↓ TNF-α BioRoot RCS > MTA Fillapex BioRoot RCS > PCS (*p* < 0.05) in hPBMC after 6 h ↓ IL-6 BioRoot RCS > AH plus (*p* < 0.05) PCS > MTA Fillapex (*p* < 0.05) in hPDLSCs after 12 h	[51]
L929 mouse fibroblast cells	Zinc oxide eugenol and methacrylate based EndoREZ sealers (ZE and ER and simvastatin incorporated sealers (ZES and ERS)	IL-6	MTT assay, Live and dead cell assay, Flow cytometry analysis	↑ IL-6 ZE > ER > ERS > ZES > Control	[52]
RAW264.7 macrophages	surface-reaction-type pre-reacted glass-ionomer (S-PRG) filler containing root canal sealer (S-PRG sealer) and Canals N	IL-1α, IL-6, TNF-α, PPARα, IL-10, p-NF-kB	MTT assay, RT-PCR, Western blotting, ELISA	↑ IL-10 LPS + S-PRG vs. LPS control (*p* < 0.05) ↑ PPARα LPS + S-PRG vs. LPS control (*p* < 0.05)	[5]

Abbreviations: ALP: alkaline phosphatase, ARG1: Arginase 1, Chil3: chil3 chitinase-like 3, COX-2: cyclooxygenase-2, CXCL: C-X-C motif chemokine ligand, GRO: growth-regulated oncogene, HIF-1α: hypoxia-inducible factor-1α, iNOS: inducible NO synthase, IL: interleukin, LOX: lysyl oxidase, IFN-γ: interferon gamma, MRC1: mannose receptor C-type 1, NO: nitric oxide, OCN: osteocalcin, PGE2: prostaglandin E2, p-NF-kB: nuclear factor kappa-light chain enhancer P, Pro-root MTA: PR-MTA, ROS: reactive oxygen, Retnla: resistin-like molecule alpha, TGF-β1:transforming growth factor beta, TNF: tumor necrosis factor, VEGF: vascular endothelial growth factor, ELISA: enzyme-linked immunoassay, RT-PCR: reverse transcription polymerase chain reaction, MNT: micronucleus test, MTT assay: (3-[4,5-dimethylthiazol-2-yl]-2,5 diphenyl tetrazolium bromide) assay.

**Table 2 dentistry-11-00054-t002:** Summary of immunomodulatory effect of endodontic sealers in vivo studies. “Yes” represented a low risk of bias, “No” represented a high risk of bias, and “Not” represented an uncertain risk. Furthermore, “Overall” indicated the percentage of low risk of bias in each study.

Animal	Material	Marker	Method	Outcomes	Ref.
18 4-week-old male Wistar rats were randomly divided into three groups	MetaSEAL and AH Plus	MHC class II, CD68, CD43.	Histology	Epoxy resin-based sealer induced the infiltration of MHC class II molecule-expressing cells, whereas 4-META-containing, methacrylate resin-based sealer elicited macrophage infiltration.	[54]
100 adult male Holtzman rats (Rattus norvegicus albinus) weighing 220 g–250 g were distributed into five groups	Root Canal sealer, MTA Plus, MTA Fillapex, AH Plus, and Endofill	IL-6, collagen	Histology and immunohistochemical analysis	The reduction in VvIC (volume density of inflammatory cells) increased with the increasing collagen in all the groups, except Endofill. MTA Plus, MTA Fillapex, and AH Plus induce regression of inflammation and formation of a fibrous capsule. MTA Plus and MTA Fillapex showed lower IL-6.	[55]
Sixteen young adult (8–10 weeks) Wistar rats, weighing 120-260 g	GuttaFlow Bioseal, GuttaFlow2 and AH Plus	\	Histology	All the sealers induced macrophage infiltrate, and GuttaFlow Bioseal had the most macrophage infiltrate. The resolution of inflammation was observed after 30 days.	[58]
Eighty Holtzman adult male rats (Rattus norvegicus albinus) were distributed into four groups containing 20 animals each	GuttaFlow Bioseal (GFB) and MTA Fillapex (MTAF)	IL-6, VEGF	Histology and immunohistochemical analysis	Up-regulation of inflammation: pro-inflammatory cytokine IL-6 increased, and VEGF increased with the tissue repair process.	[56]
Fifty adult zebrafishes (Pentagrit Research Lab, Chennai, India.)	ZnOE sealers	\	Histopathological analysis	Down-regulation of inflammation with the addition of petasin extract to ZnOE sealers.	[59]
Twenty-four young adult male Sprague–Dawley (SD) rats, aged 2–4 months and weighing 180–250 g	MTA, iRoot SP, BC Sealer HiFlow	CD163, CD206, CD86, mRNA of IL-1β, IL-6, TNF-α, IL-10	RT-PCR, flow-cytometry, immunofluorescence, and histology	Down-regulation of inflammation of BC Sealer HiFlow and iRoot SP was observed, and BC Sealer HiFlow promoted M2-like macrophage polarization in vivo.	[57]

**Table 3 dentistry-11-00054-t003:** Risk of bias of in vitro studies assessed by ‘Modified CONSORT checklist of items for reporting in vitro studies of dental materials’.

Author&Year/Item	1	2a	2b	3	4	5	6	7	8	9	10	11	12	13	14	Overall		Ref.
D. H. Lee, N. R. Kim, et al., 2007	N	Y	Y	Y	N	N	N	N	N	N	Y	N	Y	Y	N	40.00%	6	[3]
D. H. Lee, B. S. Lim, et al., 2007	N	Y	Y	Y	N	N	N	N	N	N	Y	Y	Y	N	N	40.00%	6	[4]
Y. Y. Lee et al., 2007	N	Y	Y	Y	N	N	N	N	N	N	Y	N	N	Y	N	26.67%	4	[39]
Brackett et al., 2009	N	Y	Y	Y	N	N	N	N	N	N	Y	Y	N	N	N	33.33%	5	[40]
S. T. de Oliveira Mendes et al., 2010	N	N	Y	Y	N	N	N	N	N	N	Y	N	N	N	N	20.00%	3	[41]
Brackett et al., 2011	N	Y	Y	Y	N	N	N	N	N	N	Y	Y	Y	N	N	40.00%	6	[42]
Braga et al., 2014	Y	Y	Y	Y	N	N	N	N	N	N	Y	N	Y	N	N	40.00%	6	[43]
Diomede et al., 2014	Y	Y	Y	Y	N	N	N	N	N	N	Y	N	N	Y	N	40.00%	6	[34]
Shih et al., 2014	N	Y	Y	Y	N	N	N	N	N	N	Y	N	Y	Y	N	40.00%	6	[44]
Lee et al., 2017	Y	Y	Y	Y	N	N	N	N	N	N	Y	N	Y	Y	N	46.67%	7	[33]
Martinho et al., 2018	Y	Y	Y	Y	N	N	N	N	N	N	Y	N	N	Y	N	40.00%	6	[45]
Jeanneau et al., 2019	Y	Y	Y	Y	N	N	N	N	N	N	Y	N	Y	Y	N	46.67%	7	[35]
Lee et al., 2019	Y	Y	Y	Y	N	N	N	N	N	N	Y	N	Y	Y	N	46.67%	7	[7]
Tu et al., 2019	Y	Y	Y	Y	N	N	N	N	N	N	Y	N	N	Y	N	40.00%	6	[46]
Gaudin et al., 2020	Y	Y	Y	Y	N	N	N	N	N	N	Y	N	Y	Y	N	46.67%	7	[47]
C. Jeanneau et al., 2020	Y	Y	Y	Y	N	N	N	N	N	N	Y	N	Y	Y	N	46.67%	7	[17]
Da Pedrosa et al., 2021	N	N	Y	Y	N	N	N	N	N	N	Y	Y	Y	Y	N	40.00%	6	[48]
L. A. B. da Silva et al., 2021	N	N	Y	Y	N	N	N	N	N	N	Y	Y	N	Y	N	33.33%	5	[49]
R. A. B. Da Silva et al., 2021	N	Y	Y	Y	N	N	N	N	N	N	Y	Y	Y	Y	N	46.67%	7	[50]
Pérez-Serrano et al., 2021	N	Y	Y	Y	N	N	N	N	N	N	Y	N	Y	Y	N	40.00%	6	[51]
Sharma et al., 2022	Y	Y	Y	Y	N	N	N	N	N	N	Y	N	N	Y	N	40.00%	6	[52]
H. S. S. Thein et al., 2022	N	Y	Y	Y	N	N	N	N	N	N	Y	N	Y	Y	N	40.00%	6	[5]

**Table 4 dentistry-11-00054-t004:** Risk of bias of in vivo studies assessed by ‘SYRCLEs RoB tool’.

Item/Authour&Years	Yamanaka et al., 2013, [54]	Saraiva et al., 2018, [55]	Santos et al., 2019, [58]	Delfino et al., 2020, [56]	Vinola et al., 2021, [59]	Yang et al., 2022, [57]
Selection bias Sequence generation	NO	NOT	NO	NO	NO	NOT
Selection bias Baseline characteristics	NOT	NOT	NOT	NOT	NOT	YES
Selection bias Allocation concealment	NO	NO	NO	NO	NO	NO
Performance bias Random housing	NO	YES	NO	YES	NOT	NOT
Performance bias Blinding	NO	NO	NO	NO	NO	NO
Detection bias Random outcome assessment	NO	NO	NO	NO	NO	NO
Detection bias Blinding	YES	NOT	NOT	NOT	YES	NOT
Attrition bias Incomplete outcome data	YES	YES	YES	YES	YES	YES
Reporting bias Selective outcome reporting	YES	YES	YES	YES	YES	YES
Other Other sources of bias	YES	YES	YES	YES	YES	YES
Overall	40.00%	40.00%	30.00%	40.00%	40.00%	40.00%
	4	4	3	4	4	4

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
