# Peer review of "Immunomodulatory Effects of Endodontic Sealers: A Systematic Review"

_dentistry, 2023, doi:10.3390/dj11020054_

Round 1

Reviewer 1 Report

On request of Dentistry Journal, I have revised the manuscript titled “Immunomodulatory impacts of endodontic sealers: a systematic review
However, I suggest Dentistry Journal to reconsider the paper after the authors have addressed the following:   Line 7: please delete correspondence   Line 15: in vivo or in vitro?   Line 43-45: Not only MTA, all Calcium silicate based materials, please use the following article to clarify this point: Kharouf, N.; Sauro, S.; Eid, A.; Zghal, J.; Jmal, H.; Seck, A.; Maculuso, V.; Addiego, F.; Inchingolo, F.; Affolter-Zbaraszczuk, C.; Meyer, F.; Haikel, Y.; Mancino, D. Physicochemical and Mechanical Properties of Premixed Calcium Silicate and Resin Sealers. J. Funct. Biomater. 2023, 14, 9. https://doi.org/10.3390/jfb14010009   It's important to mention the use of PRF and its advantages in the introduction to clarify the organic and biodegradable materials, you can use the following reference: Eid, A.; Mancino, D.; Rekab, M.S.; Haikel, Y.; Kharouf, N. Effectiveness of Three Agents in Pulpotomy Treatment of Permanent Molars with Incomplete Root Development: A Randomized Controlled Trial. Healthcare 2022, 10, 431. https://doi.org/10.3390/healthcare10030431   Materials and methods: please precise the period (year-year) of this review   Please through the text and figure standardize the name of the products, such as Bioroot or Bioroot RCS?....   Figure 3: Is it an original figure or copied from another paper? if it's not original, the authors should mention the original reference and have an authorization   Figure 4: the same comment for Figure 3   ZOE? please clarify and not use the abbreviation before the complete name   Reference: please follow MDPI style   The article is good but need some minor revisions

Author Response

Thank you for your letter and for the reviewers’ comments concerning our manuscript entitled “Immunomodulatory impacts of endodontic sealers: a systematic review” (dentistry-2120104). Those comments are all valuable and very helpful for revising and improving our paper, as well as the important guiding significance to our research. Revised portions are marked in yellow on the paper. The main corrections in the paper and the responses to the reviewer’s comments are shown as an attached word file.

Reviewer 2 Report

The aim of this review was to determine the effect of the endodontic sealers on the modulate of the immune system and tissue repair. Nevertheless, the manuscript appears as a sequence of half-baked ideas, where a clear vision of the effect of endodontic endodontic cements in the modulation of inflammation and tissue repair is lacking.

In the abstract section, Pro-Root MTA was included, but that Pro-Root MTA it not endodontic sealer. On the other hand, the conclusion of the study is not clearly.

In the introduction section, pulp capping materials (eg Pro root MTA or MTA) were included. However, according to the manufacturer's instructions, they are not used as endodontic sealers. On the other hand, the introduction refers to the inflammation produced by endodontic sealers in the periapical tissues, but also to the inflammation produced by pulp capping materials in the pulp tissue. This is not appropriate for the purpose of the review. Also, many topics are addressed that should only be addressed in the discussion section. Finally, the objective refers to “various endodontic materials”, which is a very broad concept that not only includes endodontic sealers. This section this section must be greatly improved according to the objective of the study.

In the Materials and Methods section, the review questions (PICOT) must be included and the PROSPERO number registration. The description of the exclusion criteria is confusing, it should be improved.

In the result section, a list of publications that were excluded after full-text search and reasons for exclusion should be added. The figures should be improved and result of capping endodontic materials must be excluded. The reference of Figure 2 and how the statistics were made is not clear.

In the discussion section, address the recommendations for future research and discuss the clinical relevance of the review results.

The conclusion of the work is very broad, it must be rewritten.

Author Response

(The authors gave the same response as above.)

Reviewer 3 Report

I was pleased to review the article entitled “Immunomodulatory impacts of endodontic sealers: a systematic review” for the Dentistry Journal. The article brings a novel and exciting topic, in addition to being well-written, with appropriate methods and figures presented. In my opinion, the manuscript is suitable for publication in its current form.

Author Response

Thank you for your letter and for the reviewers’ comments concerning our manuscript entitled “Immunomodulatory impacts of endodontic sealers: a systematic review” (dentistry-2120104). Those comments are all valuable and very helpful for revising and improving our paper, as well as the important guiding significance to our research. 

Round 2

Reviewer 2 Report

The aim of this review was to determine the effect of the endodontic sealers on the modulate of the immune system and tissue repair. Although the objective of the study is relevant, this objective is lost in the manuscript.

In the abstract section and introduction section, Pro-Root MTA was included, but that Pro-Root MTA it not endodontic sealer.In the introduction section, pulp capping materials (eg Pro root MTA or MTA) were included. However, according to the manufacturer's instructions, they are not used as endodontic sealers.

On the other hand, the conclusion of the study is not clearly. The conclusion of the work is very broad, it must be rewritten.

Author Response

Point 1: The aim of this review was to determine the effect of the endodontic sealers on the modulate of the immune system and tissue repair. Although the objective of the study is relevant, this objective is lost in the manuscript.

Response 1: We gratefully appreciate for your valuable suggestion; we modified the aim of this study to make it clear. The revised aim should be” This review aimed to answer whether endodontic sealers can modulate the immune cells and inflammation.”

Point 2: In the abstract section and introduction section, Pro-Root MTA was included, but that Pro-Root MTA it not endodontic sealer. In the introduction section, pulp capping materials (eg Pro root MTA or MTA) were included. However, according to the manufacturer's instructions, they are not used as endodontic sealers.

Response 2: Thank you for your comments. We have removed all the description of Pro- Root MTA cement in the abstract and introduction, lines 18 and lines 44 to 46.

Point 3: The conclusion of the study is not clearly. The conclusion of the work is very broad, it must be rewritten.

Response 3: Thank you for your comments. We have revised the conclusion part to address your concerns and hope that it is now clearer, lines 485 to 489.
